# New Strategies to Improve Co-Management in Enclosed Coastal Seas and Wetlands Subjected to Complex Environments: Socio-Economic Analysis Applied to an International Recovery Success Case Study after an Environmental Crisis

Salvador García-Ayllón 

Department of Civil Engineering, Technical University of Cartagena, 30009 Cartagena, Spain;
salvador.ayllon@upct.es; Tel.: +34-968325768

**Abstract:** Enclosed coastal seas and wetlands are areas of high ecological value with singular fauna and flora, but several cases of environmental catastrophes in recent decades can easily be referenced in the international literature. The management of these natural territories is complex in developed countries since they are usually subjected to intense human activity with a varied catalog of activities and anthropizing features that alter the balance of the ecosystem. In this article, the concept of the Socio-Ecological System (SES) to diagnose and achieve a sustainable cohabitation between human anthropization and the natural values based on the tool of GIS participatory mapping is proposed as an innovative approach for the management and recovery of these complex areas. The article develops a comprehensive general methodology of spatial GIS diagnosis, planning, and co-management implementation between public and private stakeholders combined with economic tools such as the Willingness to Pay (WTP) and the Cost Transfer Sector (CTS). This innovative approach is applied to the Mar Menor lagoon, which is an international and successful case study of environmental recovery on the Spanish Mediterranean coast. The coastal lagoon suffered an unprecedented eutrophication crisis in 2015, but it managed to recover in the summer of 2018 without the need to implement major structural measures. In this case study, several solutions to redress the current impacts will be developed through a participatory process based on GIS mapping. Lastly, the discussion reflects the concept of self-resilience of an ecosystem based on the unexpected positive turn of the environmental crisis in the lagoon ending.

**Keywords:** socio-ecologic system; GIS participatory mapping; environmental crisis; enclosed coastal seas and wetlands; willingness to pay; cost transfer sector; public-private co-management; diffuse anthropization

## 1. Introduction

### 1.1. The Concept of Management for the Recovery of Natural Areas after an Intense Process of Anthropization

Enclosed coastal seas and wetlands are traditionally areas of high ecological importance with valuable fauna and flora [1–3]. They provide significant ecosystem services for biodiversity such as food, recycling and removal of dangerous chemicals, climate regulation, culture and landscape, and more [4–6]. Because of these factors, their natural and geographical conditions are also usually interesting for the development of human activities such as tourism, agriculture, industry, and more [7–9]. This makes managing these maritime-coastal territories sometimes very complicated, which results in well-known

examples of environmental catastrophes caused by the impact of human anthropization in developed countries [10–13]. The recovery of these natural spaces after periods of long exposure to anthropogenic impacts or the emergence of specific environmental crises is usually very difficult.

As the first step, it is initially necessary to determine exactly the focus of the environmental problem, or which elements cause the anthropic transformation process, and diagnose its clear origin [14]. This task may not prove easy. The origin of the detected phenomenon is not always easy to determine or does not respond to a single agent but is the sum of a combination of multiple factors [15,16]. Afterward, there is the problem of implementing the necessary measures to put an end to those current impacts. This issue may give rise to numerous, rather complex, scenarios and usually creates several conflicts of interest among the stakeholders affected.

On the one hand, we have the problem of the economic cost involved in the task of recovering the affected natural area, given that the economic cost of reversing the process and restoring the area to its previous status is usually far greater than the cost of altering it [11,17]. On the other hand, the usual controversy arises from determining who should be responsible for carrying out this restoration process [18,19]. On many occasions, the fact that the administrations take on this process is very controversial because, although it does guarantee to a large extent a correct action, it fails to respect the "polluter pays" principle [20–23]. At other times, the social cost of completely eliminating the focus of anthropization is not politically acceptable since it implies ending economic activities that provide many jobs or involves an activity with strong local roots [13,24].

This context usually requires the implementation of complex management systems for the recovery process, which must be sustainable over time and able to even retain cohabitation with the presence of pre-existing human activities [25–27]. The formulas and mechanisms of these processes are not easy to standardize and require periodic updates at the research field, since the level of complexity of the impact on our society in developed countries is growing [28,29]. Moreover, the rapid incorporation of developing countries into the global production process and mass consumption has multiplied the number of cases that exist, which makes scientific research a major issue for the future [30–32].

The scientific literature provides interesting cases of major environmental disasters such as the Salton Sea (US) or the Thau lagoon (France) that have required great recovery plans. The management for the recovery of the 974 km$^2$ Salton Sea lake, which was a tourist picture postcard location in the 1950s with sandy beaches and thousands of migratory birds, has proved very controversial. Its ecosystem is currently severely damaged, mainly due to the effects of agricultural activities, which leaves it saturated with salts and pesticides [33]. The US Authorities, through the Salton Sea Land Act from 1999, planned great works for 75 years seeking the lake's rehabilitation [34]. Nevertheless, the development of such a long-term proposal, the need for very important public investment during periods of economic crisis, and the absence of involvement of many of the private process stakeholders means that, 20 years after the beginning of the plan, appreciable results are difficult to demonstrate.

The Thau coastal lagoon located in France represents another interesting case. This natural area of 70 km$^2$ of surface has been subjected to a process of diffuse anthropization for decades as a result of a varied catalog of human activities (mass tourism, agriculture, fishing, marinas, and motor boating, etc.). In recent years, these activities have led to various imbalances in the ecosystem of the lagoon, such as oligotrophication and the emergence of picocyanobacteria and a toxic dinoflagellate [35], contaminated sediment [12], and algal blooms [36]. The difficulty in determining the exact causes that generated these alterations in the quality of the water prompted the development of the DITTY EU project in 2003 [37]. A decision support system (DSS) was developed for the lagoon, which gave end-users different scenarios, according to financial, socio-economic, and environmental constraints. The different scenarios were then ranked, according to the requirements of the end-users [38]. Afterward, 15 years after the start of the project, alterations in the waters of the lagoon continue to exist. Furthermore, the causes of the agents generating those imbalances continue to be heterogeneous and sometimes even unexpected. The latest crisis forced the lagoon's closure in March 2017 due to an increase in the rates of

coliforms above the levels authorized for human health. However, the cause (which was attributed initially to problems in urban sewage systems), was the defecation of the high population of birds in the lagoon, which denotes the level of complexity in the comprehensive management of the lagoon as a whole ecosystem [39].

These two examples, as well as several others [40–42], clearly illustrate the complex management problems existing in the environmental recovery of these natural areas, and the need to deepen research in this field to achieve satisfactory results in processes of this type that frequently need to be undertaken in such areas [43]. Despite the growing scientific interest in incorporating the economic variables and the social perspective into environmental recovery processes [44–46], there are currently important gaps at the level of research in this field. In addition to the usual difficulty in defining the scope of action and the responsibility that public and private stakeholders should assume in the process [47], we must also consider the increasing complexity of finding an optimal management framework [48]. In this context, the use of a socio-ecological framework implementing economical approaches and GIS methodologies for the diagnosing and managing complex natural areas exposed to the intense process of diffuse or specific anthropization can prove very interesting [2,49,50]. The breakthrough developed by GIS tools in recent years can bring new high value approaches to the existing integrated management strategies in the field of diagnostics [14,51], participation of stakeholders, and implementation of measures for the recovery of these areas facing environmental crises. This article presents a comprehensive socio-ecological framework implemented with GIS and cost value methodologies for the Mar Menor lagoon in Spain and its latest results, which have become a case of some success in 2018 in the aftermath of an environmental crisis of the international relevance in 2015.

*1.2. The Mar Menor Case Study*

The Mar Menor is a salt lagoon of 170 km$^2$ located on the Spanish Mediterranean coast (Figure 1). It is separated from the Mediterranean Sea by an old dune strip, which allows the exchange of water only through five natural channels called golas [52]. This configuration gives it a hypersaline character that has generated a valuable ecosystem with crystal clear waters. This enables abundant marine fauna and flora to flourish, with singular species such as Pinna nobilis (the largest bivalve mollusk in the Mediterranean Sea) or autochthonous variants of the hippocampus.

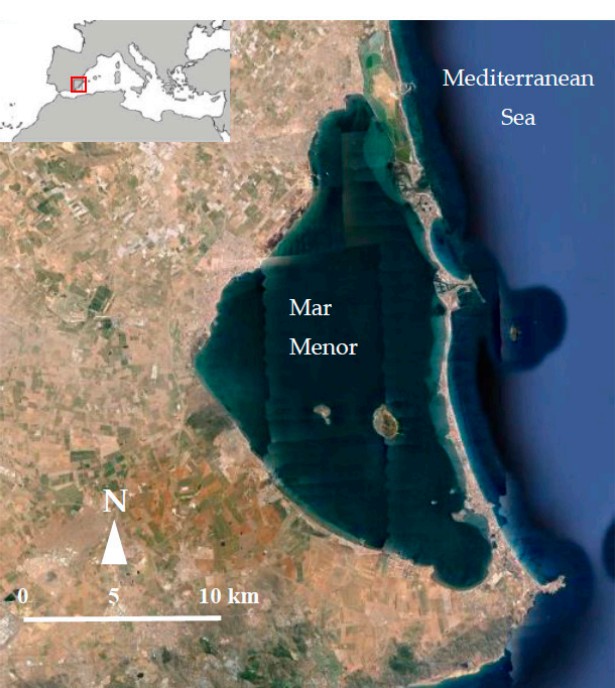

**Figure 1.** Mar Menor.

Human activity (present since the Roman era with small mining activities, fishing, or salt mines) has increased since the mid-20th century with the arrival of mass tourism. This has intensely urbanized a great deal of the coastal perimeter (the coastal perimeter has gone from being 4.5% urbanized in 1956 to the current level of 76.6% while the population grew from 15,000 inhabitants to the 600,000 reached with tourists in the summer), and recreational navigation developed with the construction of 10 marinas and the introduction of numerous motor boats [53]. From the 1980s, the construction of the water transfer network in Spain between the rivers Tajo and Segura contributed to a strong development of intensive agriculture around the lagoon [54]. From the 1990s, the Mar Menor became a natural, highly protected territory being catalogued as Special Protection Area (SPA), Special Area of Conservation (SAC), and Marine Protected Area (MPA) by the European network Natura 2000 and included as an international RAMSAR wetland, in addition to the development of other local and regional environmental protection figures.

In this period, anthropic activity along the perimeter of the lagoon began to be progressively restricted in issues such as the construction of houses, marinas, earthworks on beaches, etc. because these elements became the main issues of controversy and claims from social platforms and environmental groups. The only pending threat still to be resolved appeared to be the land drags arriving through the wadis from the nearby areas during flooding and its subsequent sediment at the bottom of the lagoon. However, the exceptional nature of a serious impact of these events, associated with the torrential rains of the Mediterranean climate during the autumn-winter and the fact that they did not affect summer tourism left this question to not be considered a priority at the social and scientific level. From that time, we also saw an exponential growth in the number of jellyfish of the species Cotylorhiza Tuberculata, whose proliferation reached its annual maximum in the summer with populations exceeding 100 million specimens [49]. This question did, however, generate greater social relevance, which forced local administrations to collect thousands of tons of these jellyfish periodically in order to not harm tourism (see Supplementary Materials).

An intense phenomenon of eutrophication in the lagoon bloomed in the summer of 2015, which transformed the traditionally crystal-clear waters into a greener color. The turbidity of the water meant that the level of visibility, always surpassing at least two meters in the lagoon, did not even reach 10 centimeters (see Supplementary Materials). This loss in transparency of the water prevented the sunlight from reaching the seabed, whose vegetation cover almost completely died in just one year. The phenomenon, which was accompanied by the disappearance of the jellyfish population, caused a major social alarm. The situation, aside from the environmental issues, generates heavy economic losses for tourism through the loss of the blue flags on all beaches since 2016.

In this context of the environmental crisis, a project for the diagnosis and implementation of solutions was launched in 2015 through the European mechanism for financing an Integrated Territorial Investment (ITI) [18]. The present research has been carried out within the structure of this project. The investigation may be of great interest for researchers in the field of environmental recovery processes, since it has developed an integrated framework for diagnosis and management in which all involved stakeholders participated. In this sense, an innovative socio-economical perspective for evaluating the viability of the recovery process is proposed. The analysis introduces an unusual point of view of a GIS participatory mapping approach to determine the stakeholders' degree of responsibility and participation in the problems and their solutions, which mixes the GIS approach with socio-economic concepts in the field of managing natural protected areas, such as the Willingness To Pay (WTP) and the Cost Transfer between Sectors (CTS). This enables a public-private partnership (PPP) framework to be established for managing the solutions in an optimized way.

The following sections will develop the methodological aspects carried out in the project. Then the results obtained for the case study will be explained. Lastly, the successful and unexpected situation will be addressed in the discussion section.

## 2. Methodology

The lagoon recovery project is proposed in two major phases (Figure 2) to implement the process and gain the commitment of the stakeholders to establish a framework of sustainable cohabitation in the future. The methodology of the two phases will be described through two subsections. The philosophy of the project is not based on a sanctioning or punitive purpose, but on an approach that enables a sustainable framework of cohabitation of the existing activities and the natural values of the lagoon to be established. In this way, the stakeholders' commitment with the solutions adopted is guaranteed, as well as the maintenance of a new framework of co-responsibility that prevents the current situation from recurring in the future. However, this does not mean that the solutions adopted do not reflect, in a fair and balanced manner, the degree of responsibility of the current environmental crisis in each of the agents involved.

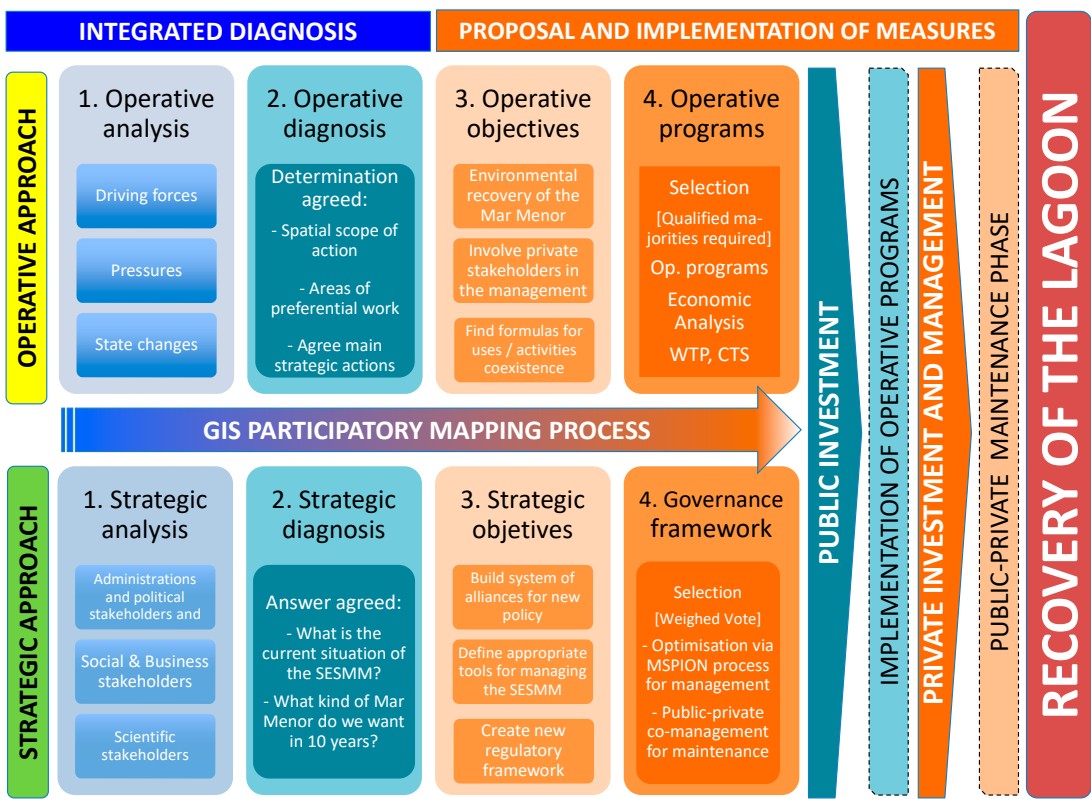

**Figure 2.** GIS participatory mapping process and co-management strategy for Mar Menor recovery.

The first phase develops an integrated diagnosis that must be able to involve all stakeholders and determines all the issues that affect the current situation of the lagoon in a hierarchical manner. The second phase focuses on the proposal of measures and their implementation in the short, medium, and long-term in a coordinated manner through the establishment of a governance framework for the Mar Menor. Both phases are, in turn, developed under two approaches that run in parallel: a more operational and a strategic approach. The backbone of this biphasic procedure is the so-called GIS participatory mapping process. This concept allows us to diagnose, analyze, and propose measures based on the objective and detailed spatial information provided by the GIS indicators. The framework facilitates reaching agreements and a technical-social consensus among the different stakeholders.

This process leads to the phase of implementation and the maintenance of measures through a system of public-private collaboration. This co-management system guarantees the minimum public investment for a comprehensive recovery of the lagoon as well as the private investment necessary to sustainably maintain the measures. In this way, the execution of the measures is optimized, which

makes the stakeholders co-responsible for the tasks for which they are more specialized. The process of implementing these different stages will be described in detail below.

## 2.1. Integrated Diagnosis

In the first place, the so-called Socio-Ecological System of the Mar Menor (hereinafter SESMM) must be configured to develop a decision panel formed by the legitimated stakeholders. Given that the situation of the Mar Menor goes beyond mere environmental problems, it is necessary to establish a framework that approaches the situation from a multidisciplinary focus in which it is possible to involve all stakeholders. Stakeholders have been selected following a DPSIR (Driving forces, Pressure, State, Impact, and Response) model [55] (Figure 3).

**DRIVING FORCES**
Tajo-Segura rivers transfer, population growth, emergence of mass tourism, old open-pit mines, salt industry, etc.

**HUMAN PRESSURE**
coastal urbanization, construction of coastal infrastructures, intensive agriculture, motor boating, fishing, diving, growth of cities, etc.

**ECOSYSTEM STATE CHANGES**
wetlands loss, landscape change, alteration in the benthos, sedimentary circulation modified, temperature and salinity, nitrates from agriculture, phosphates from urban sewage

**RESPONSES : measures to mitigate or correct dysfunctions observed**

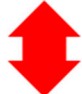

**IMPACTS**
- Alteration of beaches
- Land transformation
- 95% of lagoon seabed damaged or died
- Urban waste discharges to the lagoon
- Eutrophication of water
- Decrease of temperature and salinity of the lagoon
- Modification of species

**Figure 3.** DPSIR model developed for the diagnosis process.

During the first stage (Driving Forces), a preliminary analysis has been carried out through which the most significant elements of the problem and possible information prescribers have been determined. This analysis has allowed us, during the second phase (Pressures), to determine the stakeholders to be included in the SESMM through numerous interviews with the prescribers determined in the previous stage. The weighting of each one of the stakeholders has been obtained in the following stage (State) by measuring the degree of cross-references linked to the involvement with the causes of problems, possible solutions, management responsibility, and capacity of being objective evaluators and interested evaluators, according to the information from the interviews and following Formulas (1) and (2).

$$\varphi_i = F(\alpha, \beta, \delta, \gamma, \varepsilon) = \lambda_1 \alpha + \lambda_2 \beta + \lambda_3 \gamma + \lambda_4 \delta + \lambda_5 \varepsilon \ \text{ with } \sum_i \lambda_i = 1 \tag{1}$$

$$\Psi_i = H\left(\frac{F(\alpha, \beta, \delta, \gamma, \varepsilon)}{\sum_n \varphi_n}\right) \quad \text{ with } \Psi_i \in [0,1] \tag{2}$$

where $\varphi_i$ is the function that evaluates the number of references of prescribers for a stakeholder $i$ in relation to its responsibility in the causes $\alpha$ of the problem, in its solutions $\beta$, in its management $\delta$ at a competitive level, its capacity as an objective evaluator $\gamma$, and its capacity as an interested evaluator

$\varepsilon$. $\lambda_i$ is the importance factor of these variables and is a design parameter of the analysis. $\Psi_i$ is the function that evaluates the relationship between the number of references of this stakeholder and the number of total references.

Stakeholders must be involved in the four stages of the DPSIR model to be legitimate in order to participate in the recovery process. The development of a model based on interrelated thematic features and not a finalist evaluation based on watertight analysis allows us to prioritize the importance of the agents that must participate in the process. When configuring the decision panel of the SESMM, it will be necessary to take into account that a stakeholder may appear several times for different topics analyzed in the DPSIR model. In this way, the relative weight of each of the stakeholders will be different since each of them has different weightings in the decision process.

Once the stakeholders have been established, it is important to guide the debate properly to obtain homogeneous results that allow the establishment of a comprehensive diagnosis. To implement a truly integrated diagnosis, all the stakeholders (or at least sufficiently majority consensus) must reach agreement on issues such as the scope of action of the problem, the degree of interrelationship of each of the stakeholders with the current situation, and the role of each of the agents in the future process. Agreements will be based on consensus by establishing a qualified majority function $\Phi$ based on Formula (3), where $\Phi$ is a design variable adjusted to the specific process.

$$\Phi_j = I\left[\frac{\sum_n \Psi_n \varphi_n}{\sum_n \Psi_n}\right] \text{ with } \Phi_j \in [0,1] \tag{3}$$

where $\Phi_j$ is the qualified majority function for each decision $j$ performed by the $n$ members of the SESMM through an approval function I that is a design variable modeled for each study.

Four discussion groups including members from the four main categories of selected stakeholders (social, scientific, administration, and business categories) comprise the decision panel. The analysis is conducted following the organizational criteria of Table A1. For the development of the process, the concept of GIS participatory mapping (GPM) is implemented as an innovative approach in the field of diagnosis of issues in natural areas. This tool allows us to spatially assess the relationship and crossed links between environmental and anthropic issues by using GIS indicators. The analysis should provide detailed spatial data to reach agreements among the stakeholders about the spatial scope of action to implement the recovery of the lagoon and to hierarchize the areas of the preferential work within it.

### 2.2. Proposal of Measures, Implementation, and Management

The phase of the proposal and implementation of measures was developed based on the results obtained during the integrated diagnostic stage. This phase has been carried out following two different approaches developed in parallel. On the one hand, a strategic approach seeks to establish a new governance framework that enables the sustainable coexistence of human activities with the natural values of the lagoon to avoid new environmental crises. This framework must lay the foundations of the new regulatory context to ensure that the measures, which will be implemented during the recovery process are maintained and their management is balanced. On the other hand, an operational approach seeks to generate various programs that should face the issues detected during the diagnosis stage. These programs must develop and agglutinate the different executive actions for the recovery of the lagoon.

The solutions are proposed by stakeholders and grouped into various operational programs to differentiate those that can be implemented in the short, medium, or long-term. In this sense, it is important to combine measures from the three groups in a balanced way. On the one hand, administrations may find it difficult to address an excessive number of short-term measures economically, and may generate tensions with affected stakeholders. Conversely, an excessive number of very ambitious but long-term measures can convey to citizens the feeling that nothing is being done and it can expose the recovery plan's execution to future political swings in the administration or the

impact of inevitable economic cycles. The measures proposed were previously validated by the SESMM by means of a majority function $\Phi$ and, subsequently, selected by a function of priority $\chi$ that takes into account the importance, urgency, and motricity criteria for each proposal, according to Formula (4).

$$\Phi_j = \Phi \left[ \sum_n \Psi_n \varphi_n \right] \tag{4}$$

$$\chi_i = \chi[\rho_i, \sigma_i, \omega_i] \quad \text{with } \chi_i \in [0, 1] \tag{5}$$

where $\chi_j$ is the priority function applied to each $j$ solution selected after evaluation, which implements the weighting and correcting factors $\Psi$ and $\varphi$ applied to the vote of the $n$ members of the SESMM. The weightings given to each member of the SESMM's vote are obtained through a function that implements three characteristics of each $i$ proposal: importance $\rho$, urgency $\sigma$, and motricity $\omega$ (by motricity we refer to the ability of a proposal to generate positive inertias in other solutions proposed for this or other problems). All these three factors are specific design parameters of the analysis.

The operational programs must take into account which stakeholders should manage their execution by using a $\Omega$ matrix selective process of implementation for optimal management (MSPIOM). The matrix will select stakeholders responsible for the process by taking into account factors such as the degree of responsibility of the agent who generated the problem in which the stakeholder is the most qualified for the implementation and correct maintenance of operational programs, or what each stakeholder's competence at the administrative and legal level is. The list of possible stakeholders' participation will be crossed with different models of public-private configuration of the process by configuring the $\Omega_{ij}$ matrix. These models will calibrate private participation at different stages of the process of the lagoon recovery (i.e., main/secondary/short-term/long-term, etc. investments, management/maintenance/exploitation, etc. costs, or other features, all of which are design parameters of the specific case study).

As indicated above, while not all these questions are raised from a punitive or sanctioning point of view during the process, the approach is entirely positive to facilitate the participation and commitment of the stakeholders to the project. In this context, it must be borne in mind that, when defining possible responsibilities for the environmental crisis generated, there may be both responsibilities for actions by private agents, as well as inaction or lack of supervision by the public administrations. This legal determination may be very controversial (and not easy to determine when so many stakeholders are involved) and does not lie within the scope of this project. Clearly, the determinations from the project are not exempt from any legal responsibilities that may be derived for the environmental crisis generated and which must be settled by the pertinent judicial proceedings.

The selection of measures is made through the participatory process by applying the GIS mapping concept to clearly define the scope of each of them. This participatory process uses the same weighting factors that were finalized in the diagnostic phase by taking into account the level of importance of each of the stakeholders in the overall solution. In this case, at the level of resource allocation, the project is based on the philosophy of optimizing the financing and maintenance of the solutions proposed to recover the lagoon under a criterion of justice and responsibility ("polluter pays principle [56]"), but, at the same time, from a realistic approach. This second approach should optimize the efficiency for managing the process and ensure sustainable future cohabitation between the whole ecosystem and the pre-existing economic activities in the area (or at least a reasonable maintenance of them). To achieve this target, we will calibrate economically the level of commitment of the stakeholders involved in the $\Omega_{ij}$ matrix. This process will be performed by using the Willingness To Pay (hereinafter WTP) models for benefiting environmental ecosystems and natural tourism resources, which are based on Newton et al. [57] and Haab & McConnell [58,59].

Both non-parametric (with Turnbull estimation, [60]) and parametric (with logit/probit estimation, [61]) approaches that evaluate WTP values will be used. We calculate non-parametric Turnbull lower and upper bound estimates of WTP mean for the necessary investment for the lagoon's

recovery and its sustainable maintenance. We also model the probability of answering "yes" to the WTP question as a function of the coverage level of the payment and stakeholder characteristics. The mathematical details of the formulation implemented to compose both estimates can be seen in Appendix B.

In this context, we can also find, for instance, actions that generate a benefit to one sector may be causing damage to another sector, which both can be monetized. In this sense, to propose the economical cause-effect interrelations between sectors of the global SESMM in a balanced way, models of the Cost Transfer Sector (hereinafter CTS) are implemented to the WTP evaluation. The model is calculated with (10) based on the valuation data of Velasco et al. [62].

$$\kappa_2^1 = \kappa[\Delta B_1, \Delta L_2, mean\ WTP_1] \tag{6}$$

where $\kappa_2^1$ is the suitable CTS from sector 1 to 2 that takes into account the estimated increase in the profit $\Delta B_1$ of the activity of sector 1 due to the detrimental effect on sector 2, the estimated increase in losses $\Delta L_2$ of the activity in sector 2 as a result of activity 1, and the estimated mean willingness to pay $WTP_1$ from sector 1 to adapt its activity. The $\kappa$ function is a design parameter adaptable to the specific context of the multiparametric analysis. The whole process can be summarized in the scheme of Figure 4.

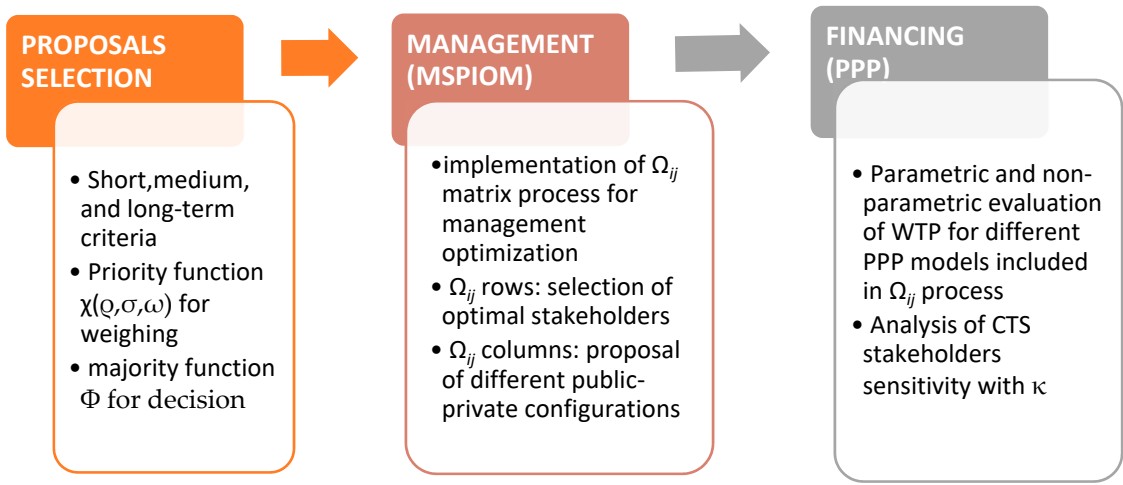

**Figure 4.** Summarized organization of the process for proposal selection, optimization management, and public-private financing evaluation for the recovery of the lagoon.

## 3. Results

Following the criteria established in the methodology section, a three-phased process (integrated diagnosis, solutions selection, and management implementation) was developed with the following results.

### 3.1. Integrated SESMM Diagnosis

The SESMM has been configured from the DPSIR model described in the previous section. This element includes all the legitimated stakeholders of the process, considering their proportional weight in the stated process by categories. The SESMM represents the decision panel through which the different stages of the global process will be completed by using the GIS participatory mapping framework. The resulting configuration of the SESMM granting the following factors of importance to the implication in the causes ($\lambda_1 = 0.25$), solutions ($\lambda_2 = 0.31$), competence management ($\lambda_3 = 0.27$), objective evaluation ($\lambda_4 = 0.23$), and interested evaluation ($\lambda_5 = 0.19$) can be seen in Table 1.

**Table 1.** Conformation of the SESMM resulting from the DPSIR analysis.

| | **Social Stakeholders** | $\Psi_i$ | **Members** |
|---|---|---|---|
| Local associations | Neighborhood associations from coastal towns, citizens platforms, etc. | 0.176 | 12 |
| Interest groups legitimized in the question | Environmental groups, associations of cultural heritage, users and traders' associations, etc. | 0.217 | 10 |
| Other social stakeholders | Sports federations (representatives from sailing, diving, jet skis, surfing, and other sports), labor unions of affected areas | 0.078 | 9 |
| | **Scientific Stakeholders** | $\Psi_i$ | **Members** |
| Universities | Professors from ecology, urban planning, biology, mining, geology, hydrology, agriculture, civil engineering, and geography departments | 0.126 | 16 |
| Scientific institutions | Research Institute for Agricultural Development and Food, Oceanographic Center of Mar Menor, Geological and Mining Institute of Spain | 0.093 | 10 |
| | **Political Stakeholders** | $\Psi_i$ | **Members** |
| Local administration | 4 Coastal municipalities + 6 others affected | 0.110 | 13 |
| Regional administration | Technical representatives from the departments of tourism, ports, environment, urban planning, agriculture, and water sewage | 0.166 | 12 |
| State administration | Technical representatives from the departments of coasts, hydrographic organizations, and water policy | 0.326 | 8 |
| | **Business Stakeholders** | $\Psi_i$ | **Members** |
| Agricultural associations | Agricultural and fishery associations | 0.338 | 12 |
| Association of tourist companies | Institutional representatives from real estate, hotel, tour operators, and hostelry associations | 0.201 | 16 |
| Other business associations | Institutional representatives from ports, nautical services companies, aquaculture, local traders' associations | 0.169 | 11 |

The members included in the four categories are distributed randomly among the four groups described in Table 1 to configure the diagnosis decision panel of the SESMM. It must be taken into account that the values of $\Psi_i$ and the number of members is not numerically correlated. The result of the $\Psi_i$ values respond only to the calculation, according to Formulas (6) and (7), while the evaluation of the number of members belonging to the solution panel is given by the qualitative interpretation of compliance with the conditions from Section 2.1 to constitute a legitimized stakeholder. Even so, we must not forget that each weighting coefficient applies to each one of the different stakeholders. Therefore, the decisions regarding the diagnosis and proposal of solutions will be affected in any case by these correction coefficients regardless of the number of stakeholders participating in the SESMM.

To develop the strategic and operational diagnoses described in Table 1, the multi-parametric analysis has been carried out using the GIS participatory mapping approach. In this stage, various agreements have been reached among the stakeholders at the strategic level to first determine the nature of the problem and what the desired future scenario should be. To this end, convergence proposals have agreed that, in the absence of hard-to-reach unanimities, they generate qualified majorities of the SESMM (in this case, the function $\Phi$ implies reaching 80% of individual weighted agreement within each of the four categories and the four groups for approval). At the strategic level, a global diagnosis has been agreed upon that assumes the main role of intensive agriculture and its contributions (surface and underground) of nitrates to the lagoon as a fundamental source of the environmental crisis that emerged in 2015 as a result of an intense process of eutrophication. Even so, the multidisciplinary nature of the process of diffuse anthropization of the Mar Menor and its surroundings, in which many other agents intervene, is also recognized. To reverse the current situation, agreement was

reached regarding the need for a governance framework that regulates pre-existing activities so that they are sustainable but from a positive approach (i.e. from a perspective of making a non-punitive regulatory framework but to control current activities by additionally implementing the necessary investments and measures to eliminate or reduce their impact so that they remain economically viable). It is important to remember that this new regulatory framework must be able to overcome the existing inadequacies of the current regulation in the territory, which suffered an important environmental crisis, despite the Mar Menor area being highly protected by various environmental figures of the Natura 2000 Network.

At the operational level, the scope of action of this new regulatory framework and preferential areas of action where the necessary measures and investments will be implemented have been agreed upon. For this, a GIS analysis was developed from a multi-parametric approach at the administrative, hydrological, geological, hydrogeological, land use, and flood risk levels. The criteria agreed upon to achieve both results through the GIS participatory mapping process have had various levels of qualified consensus. At a geographical level, it has been observed that the scope of administrative action cannot be limited to municipal scope, since several municipalities are affected, and it is not considered necessary to go beyond the regional scope. In addition, the municipalities affected are not only the coastal municipalities, since at the hydrological level the area of influence of the lagoon extends to several interior municipalities. This issue is repeated both at the geological level, as well as at the risk of flood levels and as hydrogeological level, with the last of these being the biggest determinant of the three. Hydrological and hydrogeological analyses both reveal a strong correlation with the phenomenon of nitrate contribution to the lagoon from intensive agriculture even though a consensus has not been reached on which of these routes is the predominant route.

Regarding the objective established at the operational level of establishing preferential action areas, four different areas have been differentiated through the GIS participatory mapping process. The lagoon itself is undoubtedly the critical zone. However, it is an area that already has a maximum level of environmental protection thanks to the Natura 2000 Network, which, however, failed to prevent the current environmental crisis. It is, therefore, necessary to act in the three annexed zones (coastal perimeter, Mediterranean marine area, and annexed agricultural area of influence called Campo de Cartagena), which establishs a regulatory framework of land use that protects the lagoon and implements the measures that enable the situation generated in 2015 to be reverted. The consensus agreed upon for the strategic and operational diagnosis is summarized in Table 2 and Figure 5.

*3.2. Selection of Operational Measures and Strategic Governance Framework*

Once the main objectives and the framework of the diagnostic phase had been established, the proposal phase of the measures has been developed, both at a strategic and operational level. In the operational field, the proposals validated by the SESMM through a majority function $\Phi$ (qualified majority of 50% required for each category group) are represented in Figure 6 by separating short, medium, and long-term proposals.

For the selection of the solutions among the validated proposals, the function $\chi$ was applied to the proposal with the highest weighted score in each group with the following correction factors: importance $\rho = 0.40$, urgency $\sigma = 0.35$, and motricity $\omega = 0.25$. The four categories of the previous section have been grouped into two groups in order to more clearly differentiate the behavior of the stakeholders in the proposal of measures in Figure 6. The first group, which we would call social stakeholders, basically includes those that formed the first and fourth categories of the previous section (business groups, agrarian associations, environmental groups, citizen's platforms, etc.). The second group is of a more technical character and includes public administrations as well as scientific institutions. This does not imply that the first group does not contain people with a technical background, but that the interests that legitimize the presence of these groups in the SESMM is more social in nature when compared with the interests of the second, which are more technical.

**Table 2.** GIS participatory mapping criteria and Φ reached in the strategic and operative diagnosis.

| Strategic Diagnosis | Criteria for Assessing Problems and Solutions | Φ Reached |
|---|---|---|
| Current situation | - Main role of intensive agriculture in eutrophication because of nitrates contribution but existence of a diffuse anthropization impact involving other sectors | 82.8% |
| Target scenario | - Need for a new framework that regulates pre-existing activities so that they are sustainable and implement measures but from a positive approach | 86.7% |
| | - Need for the creation of an entity that ensures the coordination of all administrations with competencies in the SESMM and allows social participation | 96.2% |
| **Operative diagnosis** | **Criteria for evaluating the scope of action and areas of preferential work** | **Φ Reached** |
| Administrative analysis | - Insufficient to establish the municipal or below-regional scope as a limit since there are several municipalities involved | 97.2% |
| | - Adequate to establish as a limit of the regional scope for a better competency framework adaptation | 88.8% |
| | - Not necessary to expand to a free or broader area in order to not involve other administrations with little political attachment to the Mar Menor | 80.5% |
| Hydrological analysis | - Great importance of linking the hydrological dimension with the contribution of nitrates | 91.3% |
| | - Need to incorporate the entire watershed due to surface flows arriving to the lagoon | 77.4% |
| | - Need to incorporate a broader surface area than the watershed due to unknown/uncontrolled surface flows | 44.6% |
| Geological analysis | - The geological configuration is an important criterion to determine the scope of action due to erosion, sediment transport, mining contributions, etc. | 43.9% |
| | - The geological dimension is not a determining criterion in the configuration of the scope of action | 67.6% |
| Hydrogeological analysis | - The hydrogeological flow is a factor of great importance in the configuration of the field of action because of the contribution of nitrates | 90.9% |
| | - The hydrogeological criteria should prevail over the hydrological ones because the underground flows are greater than the surface ones | 47.1% |
| Land use analysis | - Coastal urbanization has influence on the current problem due to the contribution of phosphates to the lagoon | 58.2% |
| | - Ports, coastal infrastructure, and motor navigation have an important influence in eutrophication | 44.0% |
| Coastal flooding risk analysis | - Flooding risk configuration in the area generates important contributions of sediments to the lagoon | 93.1% |
| | - The contributions of sediments from the floods are a determining factor in the eutrophication | 46.6% |
| Areas of preferential work | - Four areas for preferential work are determined: the lagoon and three annexed areas | 87.8% |
| | - A new regulatory framework must be specifically focused in annexed areas | 74.3% |

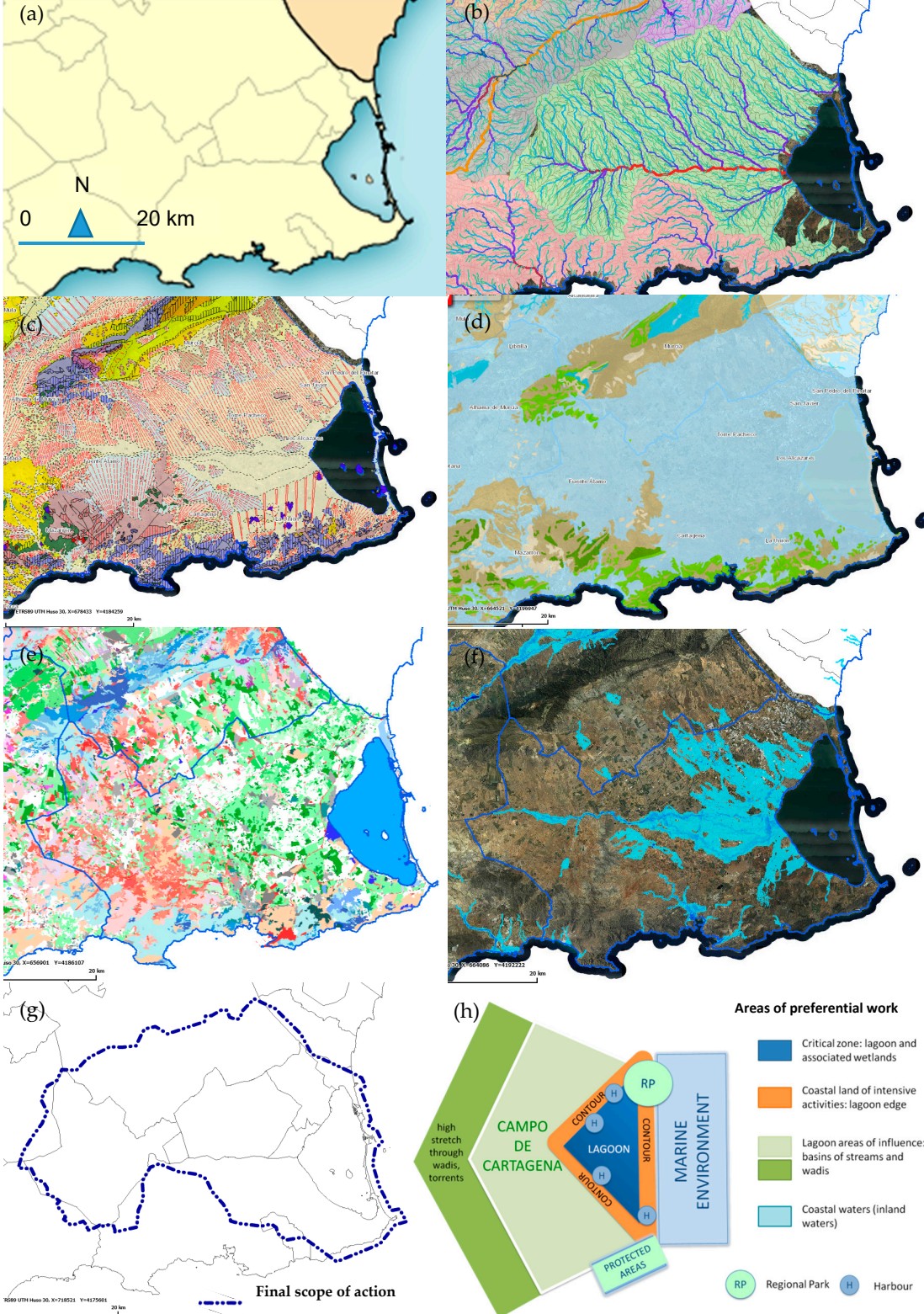

**Figure 5.** GIS participatory mapping approach: (**a**) Administrative delimitation in the area, (**b**) hydrological analysis, (**c**) geological analysis, (**d**) hydrogeological analysis, (**e**) land use analysis, (**f**) flooding risk analysis, to define the (**g**) final scope of action as the selected overlapping administrative context, and (**h**) the four preferential areas of work (see Supplementary GIS material online for more detail).

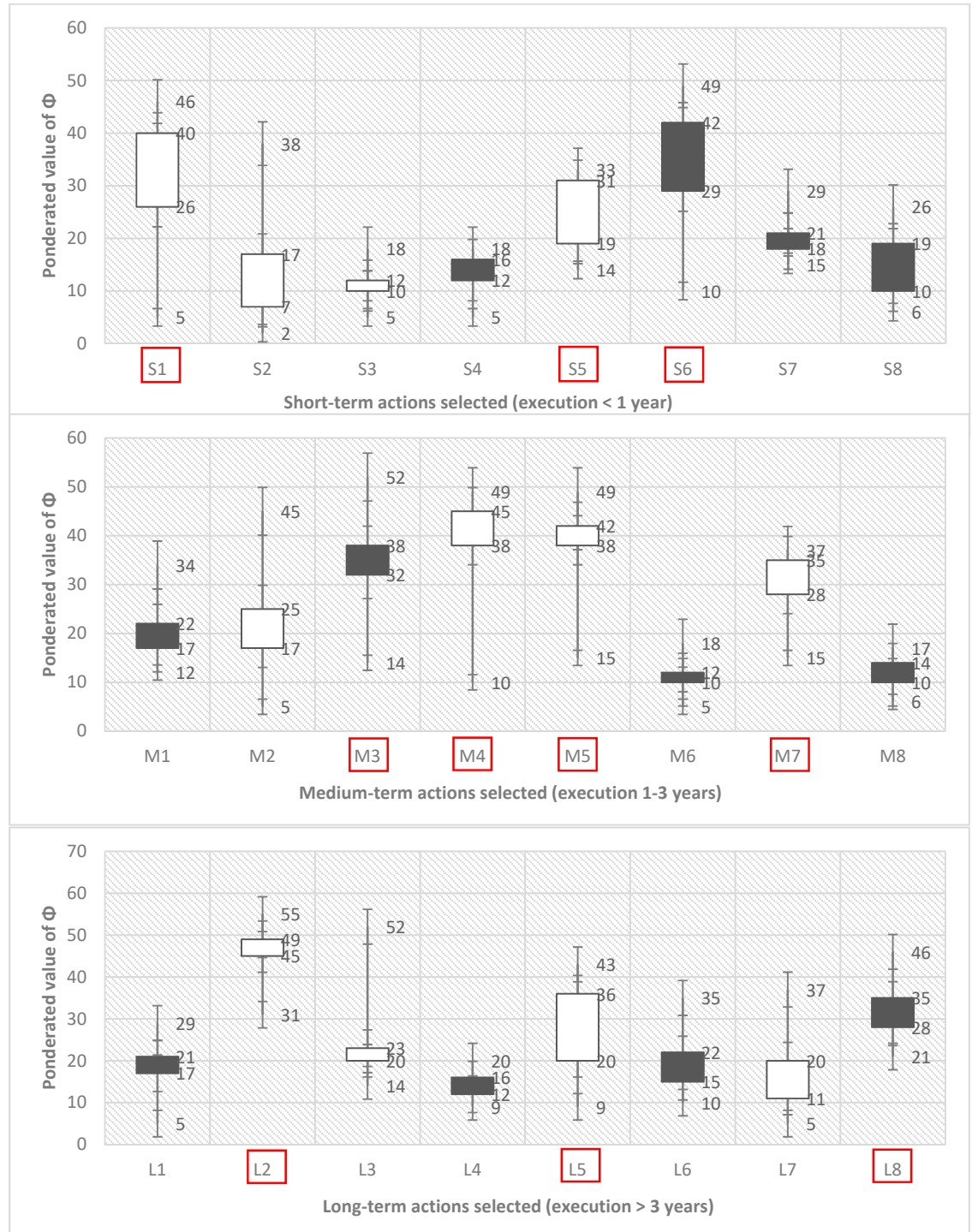

**Figure 6.** Actions selected by the SESMM (in red) including the average value granted by the "social" and "scientific" stakeholders (upper and bottom line of each box), and the maximum and minimum values obtained for each of the proposals. Note: black box when the average value of social/business stakeholders is higher than that of scientific/administrative stakeholders, and the white box is otherwise.

The operative actions that attract the greatest consensus of the stakeholders are the need to implement a plan of "zero discharges" to the lagoon (L2 in Figure 6), the implementation of urban planning actions to reduce the impacts of flooding (M3), the renovation of coastal infrastructures in a way that generates less impact on the sedimentary dynamics (L5), and the control of the underground

flow of nitrates (S1) to the lagoon. Other actions such as the construction of storm tanks to reduce the contribution of phosphates and fats from the washing of streets and urban areas (M4), the execution of natural barriers in private agricultural areas for the decrease of sediment contributions in floods (S5) or the construction of the so-called "green filters" (M7, large areas of lagooning for natural filtering of surface runoff) obtained a significantly lower consensus. Among the actions not selected since they did not fulfill the requirements established in function Φ, we can highlight the expansion and dredging of the communication channels between the Mar Menor and the Mediterranean Sea to facilitate the renewal of the waters of the lagoon (L1) due to the risk involved for the balance of the ecosystem. Similarly, imposing a tourist tax destined for the conservation of the lagoon (S2) was rejected because of the negative impact it would have on tourism (such as conservation taxes S3 and S4, which were also deemed inadequate measures). We must bear in mind that some of these actions, such as the so-called "Zero Discharge Plan", involve short, medium, and long-term actions of a structural and developmental nature (for example, the construction of a large collector that gathers all the contributions from agriculture and takes them to an authorized discharge point for treatment (S6), must be complemented by a highly branched secondary network that reaches all the large plots of intensive nitrogen agriculture, included in M4, M5, and L8). The optimal implementation, management, and cost sharing of these solutions and their analysis as main or secondary investments will be addressed in the following section using the WTP method approach.

It is interesting to observe how the social and political stakeholders fundamentally opt for more aggressive and fundamentally prompt simple and short-term solutions, while those of the latter are more oriented to proposals that are structural, complex, and focused towards the medium and long term. This question will be addressed more thoroughly in the discussion section. In reference to this question, we must also bear in mind that the separation and differentiation of the proposals into the short, medium, and long-term refers exclusively to their implementation. In this sense, a solution established as short term does not necessarily mean that its results will be observed in the short term (although it will clearly be easier for results to be observed if the execution period of a solution is shorter).

At the strategic level, as concluded in the diagnostic phase, it was agreed that a new governance global framework should be developed by creating an entity that ensures the coordination between different administrations involved in the SESMM and the social participation. This new organizational structure should also count on the scientific support of different stakeholders included in the SESMM. This new structure will be responsible for overseeing the implementation and maintenance of strategic and operational measures.

Moreover, in line with the determinations of the diagnostic phase, a new regulatory framework has been implemented for each of the four preferential work areas determined in the final scope of action. This specific regulatory framework will be developed at the technical detail level by the new inter-administrative entity with the social participation of the SESMM members and the technical support of the Mar Menor scientific committee. However, the guidelines of this new regulatory framework have been agreed upon within the SESMM by implementing the GIS mapping participatory process. In this way, a subsequent conflict at the socio-political level is avoided, and the risk of litigation at the legal level because of particular stakeholders is limited (both factors could be a posterior delay of the implementation of the regulatory framework and, therefore, the effectiveness of the comprehensive recovery plan for the lagoon). For the approval of the guidelines, Φ functions of a qualified majority have been used (60% agreement for each of the four groups in Table 2).

As the most significant example of the results obtained in the implementation of the regulatory framework in the preferential work areas, the guidelines of the regulatory framework of the lagoon's agricultural area of influence are included as supplementary material. Through the GIS mapping participatory process, three zones with different levels of restriction in agricultural use have been established: the first closest zone with a high level of restriction on agricultural use, a second with medium restriction levels, and a third one further away where merely precautionary measures are

imposed on agricultural activity to avoid the surface and underground nitrate contributions. These measures are not only focused on the agricultural sector. They also seek, for example, to stop the erosion that facilitates the surface flows of earth that settle in the lagoon after floods, or the arrival of heavy metals coming from the disused mines located to the south of the lagoon or regulate urban planning development. The result of the GIS participatory mapping process for this part of the regulatory framework can be seen spatially summarized in Figure 7.

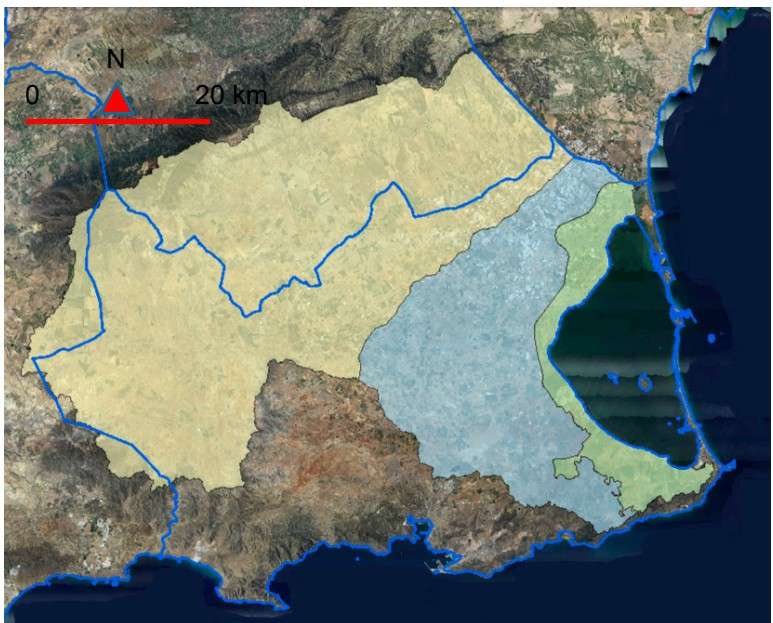

**Figure 7.** Regulation framework established for the agricultural area annexed to the lagoon with three levels of regulation: area 1 with a high level of restrictions on agricultural use (zone in green), area 2 with medium restrictions (blue zone), and area 3 with low restrictions (yellow zone).

### 3.3. Implementation and Management of the Process Applying MSPIOM, WTP, and CTS Methods

To determine who should implement, manage, and maintain the different solutions selected in the previous section, the parametric and non-parametric MSPIOM process described in the methodology section has been applied. We have introduced four different models in the simulation scenarios by taking into account five variables (Appendix C). The models propose two extreme scenarios (investment and management/maintenance of solutions fundamentally public or private) and two mixed scenarios (one with greater public participation and another with greater private participation). Within the extreme models, it must be noted that 100% purely public or private investment and management/maintenance scenarios have not been considered, since these situations would not be possible in practice due to legal or technical issues.

The sensitivity of the stakeholders included in $\Omega$ to participate in the different investment and management models for the recovery of the lagoon has been evaluated through the parametric and non-parametric estimation of Willingness To Pay. For this purpose, the level of incidence in each one of the sets of short, medium, and long-term solutions selected in the previous section has been previously analyzed at a statistical level. In this part of the study, in order to have a critical mass at the statistical level, stakeholders were asked to provide a series of opinion data from their associates (since most of them were representatives of associations) through surveys.

To estimate the upper and lower thresholds of private participation in the cost of recovery, the mean WTP value of the set of $\Omega$ was evaluated by the non-parametric Turnbull method. Subsequently, the results obtained were also corrected by implementing the sensitivity level of the stakeholders to introduce the CTS variable. On the other hand, to assess the sensitivity of WTP for the different management configurations proposed, we model the probability of a "yes" response using parametric

probit and logit specifications. The four models were tested based on two combinations of covariates, which are the solutions selected with more economic implications and the different groups of stakeholders. The results obtained are summarized in Table 3.

**Table 3.** Turnbull non-parametric and logit-probit parametric estimation of WTP of stakeholders.

| Statistical Incidence of Solutions Selected on Stakeholders from $\Omega_{ij}$ Matrix (%) | |
|---|---|
| Agricultural owners | 0.59 |
| Real estate and tourist companies | 0.23 |
| Ports and nautical companies | 0.11 |
| Other service providers related to tourism | 0.15 |
| Local administration | 0.26 |
| Regional administration | 0.39 |
| State administration | 0.47 |
| Other stakeholders with a legitimized economical interest or impact | 0.08 |

| Non-Parametric Analysis: Turnbull Estimations of WTP in % (Protest Zeros Included in the Sample) | | | | |
|---|---|---|---|---|
| Bid % ($t_{ij}$) | $F_{ij}$ ($N_{ij}/T_{ij}$) % "no" | $F_{i,j+1} - F_{i,j}$ | Mean WTP with 95% confidence | $\kappa_2^1$ |
| 0% | 0.153 | 0.153 | 84.7 ** (91.2, 78.7) | 0.09 |
| 25% | 0.218 | 0.065 | 78.2 ** (86.7, 71.4) | 0.06 |
| 50% | 0.619 | 0.401 | 38.1 ** (48.4, 32.9) | 0.02 |
| 75% | 0.929 | 0.310 | 7.1 ** (14.4, 3.5) | < 0.01 |
| 100% | 0.991 | 0.062 | 0.9 ** (6.7, 0.2) | < 0.01 |
| $E_{LB}$ (WTP) | 25.6 | $E_{LB}$ (WTP, including CTS) | | 23.4 |
| $E_{UB}$ (WTP for $T_{M+1}$=25%) | 44.2 | $E_{UB}$ (WTP including CTS for $T_{M+1}$=25%) | | 41.8 |

| Parametric Analysis: Estimated Logit Coefficients for the Probability of a "Yes" Answer to WTP | | | | |
|---|---|---|---|---|
| Solutions with greater impact | Coefficient (std. error) | | | |
| | Model 1 | Model 2 | Model 3 | Model 4 |
| Main Zero discharge Plan (Z.D.P.) | 0.527 (0.07) * | 0.415 (0.07) * | 0.202 (0.07) * | −0.067 (0.06) * |
| Green filters | 0.061 (0.05) * | −0.137 (0.05) * | −0.456 (0.04) * | −0.655 (0.04) * |
| Main Flood Plan | 0.238 (0.06) * | 0.089 (0.02) * | −0.192 (0.03) * | −0.351 (0.01) * |
| Underground flows control | 0.210 (0.06) * | 0.225 (0.02) * | 0.107 (0.03) * | −0.122 (0.01) * |
| Secondary actions Z.D.P. | 0.031 (0.05) * | 0.337 (0.02) * | 0.456 (0.03) * | 0.152 (0.01) * |
| Stakeholders | Coefficient (standard error) | | | |
| | Model 1 | Model 2 | Model 3 | Model 4 |
| Agricultural associations | 0.031 (0.02) *** | 0.437 (0.02) *** | 0.126 (0.02) *** | −0.114 (0.02) *** |
| Real estate and tourist companies | 0.235 (0.03) *** | 0.227 (0.03) *** | 0.015 (0.03) *** | −0.289 (0.03) *** |
| Ports and nautical companies | 0.354 (0.08) *** | 0.185 (0.08) *** | −0.086 (0.08) *** | −0.362 (0.09) *** |
| Other service providers related to tourism | 0.284 (0.06) *** | 0.211 (0.06) *** | 0.003 (0.06) *** | −0.318 (0.06) *** |
| Public administrations | −0.157 (0.12) *** | 0.232 (0.12) *** | 0.420 (0.13) *** | 0.036 (0.13) *** |
| Other stakeholders of $\Omega$ | 0.042 (0.07) *** | 0.227 (0.07) *** | 0.106 (0.07) *** | −0.184 (0.06) *** |
| Rest of stakeholders of SESMM | 0.033 (0.03) *** | 0.122 (0.03) *** | 0.366 (0.03) *** | 0.158 (0.03) *** |
| Online public survey | 0.084 (0.01) *** | 0.105 (0.01) *** | 0.201 (0.01) *** | 0.186 (0.01) *** |

*, **, and *** indicates statistical significance at 1%, 5%, and 10%, respectively.

If we observe the results for the different models and the possible combinations of stakeholder participation while taking into account their impact on the solutions and their predisposition to participate, we can extract interesting considerations. It is evident that, in the vast majority of cases, there is a significant aversion on the part of private stakeholders to participate in the main investments to be made in the short-term. However, there is also a positive appreciation of an important willingness to participate in the management and maintenance of the solutions implemented by these same stakeholders, and a much lower aversion to contributing to the implementation of more secondary investments.

From the point of view of different types of stakeholders, we can observe that fundamentally the agricultural stakeholders, among those economically involved, present the greatest predisposition to participate in the process. This is quite reasonable given their important link with the causes of the main problem, but also because of the economic impact that a more restrictive regulation could have on the economic viability of their agricultural activity. There is also a strong refusal to recognize and implement factors of rebalancing between sectors by costs-benefit transfer, finding only a certain

predisposition on the part of the agricultural sector to recognize some damage to the tourism sector. This context is consistent with the strong refusal to implement direct or indirect permanent taxes for the conservation of the lagoon. As expected, a greater predisposition can also be observed among the agents not affected or less economically involved in the matter (social agents or the general public) and among those most affected (business agents), which highlights the importance to demarcate this aspect in this type of participatory processes to shape realistic recovery strategies in the long-term.

In the context of efficiency in management, it should be noted that the most extreme models (1 and 4) are those that globally obtain the worst scores. It is also interesting to observe how it is more efficient for the structuring actions of the process to remain within a public management framework and, thus, avoid possible conflicts of interest between those responsible for implementing the solutions and the causes of the problems observed.

Therefore, if we focus on efficiency criteria and the predisposition to participate by different stakeholders, it is clear that the optimal solution environment would have an initial investment of the main actions (Main Infrastructures of the Zero Discharge Plan, Plan against flooding, underground run-off collection network, etc.) of a mostly or completely public character to a secondary investment (surface drainage networks in agricultural spaces, pipelines to authorized discharge points, less aggressive coastal infrastructures with sedimentary dynamics, etc.). This comes with a private majority component and a mixed public-private management and maintenance framework in the main investments and mostly private in the secondary ones.

## 4. Discussion and Conclusions

The work carried out demonstrates the importance of three common issues in environmental crises in territories subjected to processes of diffuse anthropization. In the first place, it is important to correctly delineate the cause or causes that lead to environmental problems and identify the stakeholders related to them. The origin of the problems may not necessarily be in direct contact with the affected environment - or even geographically close - as has been seen. In addition, there may not be a single focus to the problems, but rather we can find what we refer to as a phenomenon of "diffuse anthropization" (human impact on the environment in which there is no clear cause-effect relationship). This question is increasingly addressed in recent scientific works in other areas [37,63–65]. In the present case, it is evident that intensive agriculture with its contribution of nitrates to the lagoon was the main agent of the eutrophication phenomenon that led to the 2015 environmental crisis. However, the problems of the Mar Menor go beyond the surface nitrate inputs of agriculture.

In this context, we find the second issue, which must usually be addressed in this type of situation: the proposal for a solution for environmental recovery. In this context, such complex problems cannot be approached from the merely traditional scientific dimension, but the social factor, administrative viability, and economic impact to the existing productive fabric must all be considered. In this sense, the socio-ecologic system (SES) represents an innovative approach to develop comprehensive solutions to this kind of environmental problems, which has recently begun to be consolidated in the scientific context [66,67]. In the case studied, this methodology represents a significant advance toward multi-disciplinarily in approaching the problem with respect to the traditional scientific analysis previously carried out in the Mar Menor. All these studies oriented solutions in a more segmented manner to watertight proposals in the field of ecology, biology, geology, or chemistry [68–71]. In addition, the development of a complete model with majority functions $\Phi$ and priority functions $\chi$ to establish a fair and balanced framework in decision-making supposes an important evolution in the field of traditional open participatory processes. In these processes, issues such as the level of responsibility in the problems, the degree of knowledge of them or the legitimacy of each one of the stakeholders are not usually addressed and evaluated scientifically.

In line with this new approach, the development of methodologies to optimize the management of the proposed solutions and ensure the commitment of the stakeholders involved with environmental recovery strategies is essential. In this sense, the incorporation of evaluation methods for the implementation

of a public-private partnership (PPP) in the development and management of environmental solutions is a significant advance in this field. The development of PPPs in these situations is both mandatory to preserve the criterion of justice (polluter pays principle) and necessary to develop an optimized management model that assigns the responsibility best to perform in each case to each agent of the process. In addition, one of the main problems detected in natural areas affected by similar environmental crises is usually the difficulty in implementing solutions that are economically realistic in the medium and long-term. In the Mar Menor case study, the implementation of the process optimization matrix $\Omega$ and the evaluation through parametric and non-parametric systems of the stakeholders' WTP is a very innovative approach in the environmental field.

The results obtained have enabled us to know the degree of involvement of the different stakeholders in each of the solutions and their thresholds of participation in the economic section. This question is especially relevant if we seek to develop a realistic PPP process that allows us to recover the lagoon environmentally and maintain sustainable cohabitation with the pre-existing economic activities. The methodology used at the WTP parametric level has allowed us to propose four scenarios offering differing degrees of public-private collaboration. The results obtained confirm that both those models in which the great totality of the weight of the process falls on the public administration and those that fall on the private sector are the least realistic and viable models. The range of maximum thresholds of WTP obtained with the non-parametric Turnbull method allows us to assume the two mixed PPP models proposed in the parametric method are viable. These results are consistent with those obtained in similar problems in other cases of major environmental crises in developed countries [10,37], but they do present differences that may be understood as advances in the field for smaller cases [72,73] or in countries with lower environmental demands [74].

Another different question is the variant of the CTS implemented in the WTP as a rebalancing mechanism between private stakeholders. The results obtained are scarce, due to the scarce predisposition of the stakeholders to implement this type of economic rebalancing mechanisms. Only agricultural stakeholders have a certain predisposition to recognize some transfer of costs between sectors since their activity may harm the tourism sector. We, therefore, find ourselves in a context that is still too immature to deal with this issue decisively. However, this question of the indirect transfer of costs between sectors will sooner or later become a variable to be addressed in the design of econometric models that are capable of valuing complex environmental contexts by taking into account the "Beneficiary Pays Principle" in a realistic way. In this sense, the more in-depth development of the CTS variable can be an interesting future line of research.

Lastly, we must reflect on what has happened in the lagoon during recent months. This phase of starting the implementation of measures through public-private co-management coincided with the unexpected anticipated recovery of the lagoon in the summer of 2018, without having to actually implement truly structural measures. There are currently clear symptoms of recovery in the lagoon, which reach the levels of visibility prior to the 2015 eutrophication crisis. Nevertheless, the fragility of this ecosystem and the uncertainty regarding the fact that this recovery has occurred without the material implementation of structural measures opens new questions. This new situation has forced us to rethink the analysis of the project and reconsider the importance of concepts such as the intrinsic resilience of natural areas and the fragility of a recovery process over time.

The waters of the Mar Menor regained during 2018 show the levels of visibility and transparency of more than four meters in many areas. As can be observed in Figure 8, the reduction of the level of turbidity during the second half of 2018 has allowed us to recover a part of the natural marine seabed killed by the environmental crisis. It is, thus, evident that the process had not reached the "point of no return." The current context is very likely a consequence of the cessation of discharges (legal and illegal) that intensive agriculture had been carrying out for years into the lagoon. This situation was due to the insufficient (or inefficient) regulatory framework in the area, which had focused its measures on activities (such as coastal urbanization), located within the area protected by the Natura 2000 Network, underestimating the effect of other activities that were found beyond its boundaries (such as

the effects of agriculture). Therefore, the recovery is derived to a large extent by the multidisciplinary and integrated diagnosis made and the social pressure placed on agriculture during the last three years within the framework of the SESMM. The strong proliferation of the population of jellyfish *Cotylorhiza tuberculata* during the years prior to the eutrophic crisis of 2015 due to the contribution of nutrients to the lagoon from agriculture was the visible symptom of a problem that had been poorly addressed due to the absence of a correct diagnosis.

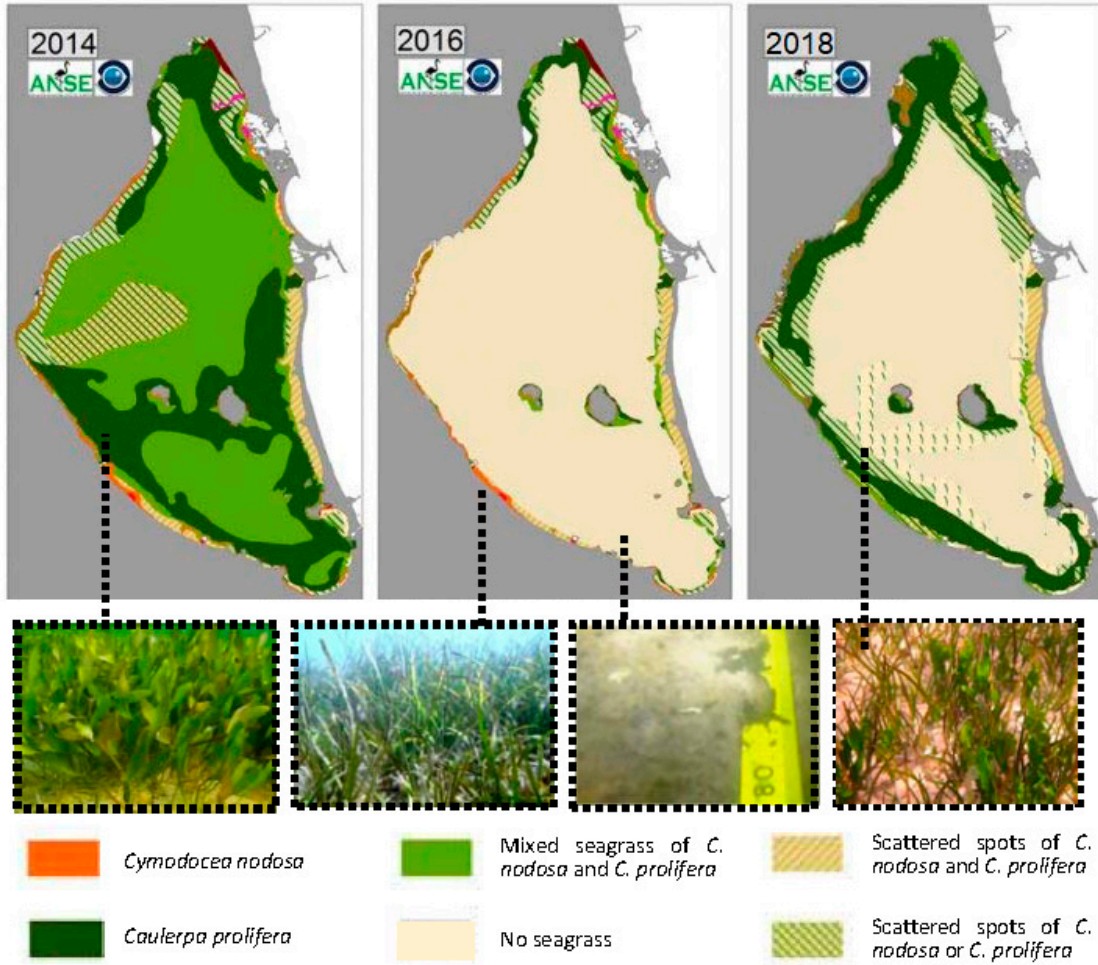

**Figure 8.** Evolution of the plant cover of the seabed of the Mar Menor (2014-2016-2018). Source: Spanish Institute of Oceanography and environmental group ANSE.

Despite the current good news, as can be seen in Figure 6, the speed of seabed regeneration is much slower than the speed of its destruction, which can give us an idea of the evolution of the global process. Even so, this new situation has raised some doubts in the process of integral environmental recovery of the lagoon. In this sense, it is, therefore, important in long-term recovery processes to factor in the environmental resilience variable of the damaged ecosystem itself. There have been some approaches to this question in the case of coastal lagoons and wetlands, such as in Reference [51]. Nevertheless, this question is currently a field in which future lines of research should be developed to refine the models and make them even more realistic.

In any case, given the lack of better and more advanced elements of judgment at present, it is evident that the possible conjunctural variation of the global context must not imply a reduction in the determination of the stakeholders to participate in the process of environmental recovery nor a reduction in the implementation of the scientifically established measures to solve the problems diagnosed. In the spring of 2017, some improvement in water transparency parameters was also seen.

However, this situation proved conjunctural because, with the sharp rise in temperatures that the region had in the summer of that year, the waters again lost all visibility, which revert to the green tonality. The present recovery seems to be far more stable. Even so, it is important to implement the set of proposed measures to permanently reverse the situation because agriculture is neither the only cause of the phenomenon of diffuse anthropization nor can the final stability of the process of environmental recovery of the lagoon be circumscribed to the simple reduction of water turbidity.

As a conclusion to all the above, we must highlight the main ideas achieved that allow the implementation of "new strategies" for the recovery of this type of natural space in other areas of the world in the aftermath of an environmental crisis. As has been observed, it is essential to correctly diagnose the problem if we aim to propose the correct solution. For this, the diagnosis must be multidisciplinary, and address the problem from both the point of view of a focalized origin, and from the perspective of diffuse anthropization phenomena without a concrete origin. In this field, the use of GIS tools combined with a socio-ecological approach can be essential to establish a framework that addresses the problem from a comprehensive perspective. On the other hand, the implementation of the solutions cannot be only approached from the scientific point of view, nor implemented only from the public administration. A socio-ecological balanced framework that is capable of integrating all stakeholders to obtain consensus solutions must be established for the diagnosis of problems, the proposal of solutions, and the management of such solutions. For this, it is necessary to approach the process from a non-punitive perspective while respecting the "polluter pays" and "the beneficiary pays" principles of sustainability. At this point, the combination of the GIS tools, a socio-ecological framework, and socio-economic analysis methods such as the WTP and the CTS has proved to be very useful in obtaining proposals that are rigorous, viable, and optimized in their management.

**Supplementary Materials:** The following are available online at http://www.mdpi.com/2071-1050/11/4/1039/s1, a graphic dossier and GIS delimitations of the final scope area of the SESMM, the areas of preferential work, and the agricultural regulatory zoning.

**Funding:** This research received no public or external funding to preserve its independence.

**Acknowledgments:** This work gathers the experience of the author as the coordinator of the European Integrated Territorial Investment project for the Mar Menor and the events and results observed during the last months of 2018. The author would like to thank the institutional support for the project of the Government of the Region of Murcia through its president P.A. Sanchez and its regional minister F. Bernabé, the collaboration of officials, technical teams, and experts of renowned prestige such as J.M. Barragán and A. Perez-Ruzafa, and the participation and commitment with the project of all the stakeholders such as local and state administrations, citizens platforms, business associations, scientific institutions, or environmental groups.

**Conflicts of Interest:** The authors declare no conflict of interest.

## Appendix A. Technical Data of the Survey Process Carried Out during the Different Phases

- Individual interviews to configure SESMM: 64 interviews [November/December 2015]
- Integrated diagnosis workshop: 100 stakeholders invited/104 attendees [February 2016]
- Proposals workshop: 100 stakeholders invited/106 attendees [April 2016]
- Final number of members involved during the process [September 2015–March 2017]: 1216 (including 1 coordinator/9 political posts from government/2 consulting teams made up of 7 people for the assembly of the workshops and the elaboration of the documents/2 scientific experts for advice of the political balanced scorecard/5 regional government officials for documents overview/147 different public and private stakeholders' representatives)
- Participants through the web portal of citizen participation and transparency: 741 (citizens participated by filling out different surveys to comment on the results and implementation of the process [April–September–November 2016]
- Public exposure of documents for claims in the official bulletin: 124 claims of private individuals filed during the period of public exposure for drawing up the final document [December 2016, 124 vs. 5472 of last environmental tool of Mar Menor exposed without SESMM participatory process].

**Table A1.** Organization of the analysis panel performed in the SESMM for the diagnosis.

| Diagnosis Approach | Topics Discussed | Target Results |
|---|---|---|
| Strategic diagnosis | Group 1:<br>- Environmental crisis<br>- Socioeconomic activities<br>- Diffuse anthropization | Reach consensus about what the current main problems of the Mar Menor are |
| | Group 2:<br>- Public policies and governance<br>- Administrative coordination<br>- Public<br>-private cooperation | Reach agreement about what scenario in the SESMM we want for the future |
| Operative diagnosis | Group 3:<br>- Driving forces: the levers of change<br>- Pressures: how human activities are affecting the whole ecosystem<br>- Environmental changes: visible problems | Reach consensus about the spatial scope of action for the SESMM and the implementation of measures |
| | Group 4:<br>- Hierarchize existing problems by levels of urgency and importance<br>- Basis for integrated management assigned within the stakeholders | Reach consensus about the hierarchization of the areas of preferential work within the SESMM |

## Appendix B. Mathematical Formulation of Non-Parametric Turnbull and Parametric Logit/Probit Estimators of WTP

The Turnbull estimator of willingness to pay presumes that, if a stakeholder answers "yes" to a particular invest or maintenance value, then we can accept that their maximum willingness to pay is at least that value. A "no" response indicates a maximum willingness to pay less than the indicated value. The lower bound (conservative) estimate of mean willingness to pay is calculated by using the equation below.

$$E_{LB}(WTP) = \sum_{i,j=0}^{M} t_{i,j} \cdot \left(F_{i+1,j+1} - F_{i,j}\right) \tag{A1}$$

where $i$ and $j$ are the indexes of the invest and maintenance amounts, $t_{i,j}$, M is the maximum value amount, and $F_{i,j}$ is the proportion of respondents who faced a particular amount and answered "no". $F_{i,j}$ represents the probability that a randomly chosen stakeholder will say "no" to amount $t_{i,j}$. The term in brackets, $F_{i+1,j+1} - F_{i,j}$, is, therefore, the difference between the proportion of "no" responses at a particular amount and the proportion of "no" responses at the next lowest amount, and is a consistent estimate of the probability that WTP lies between $t_{i,j}$ and $t_{i+1,j+1}$. In the same way, an upper bound estimate of mean WTP can, therefore, be calculated using the next-highest level of coverage $t_{M+1}$ for the investment and maintenance value.

$$E_{UB}(WTP) = \sum_{i,j=0}^{M} t_{i+1,j+1} \cdot \left(F_{i+1,j+1} - F_{i,j}\right) \tag{A2}$$

A stakeholder $i$ can be expected to answer "yes" to a particular amount for maintenance of the environmental recovery $t_j$ if their utility (satisfaction) $u_i$ with the fee is higher than utility in the absence of the fee for this environmental recovery and maintenance.

$$u_{1i}\left(y_i - t_j, X_i, M_1, \varepsilon_{1i}\right) \geq u_{0i}\left(y_i, X_i, M_0, \varepsilon_{0i}\right) \tag{A3}$$

where $u_i$, $y_i$, and $X_i$ represent the stakeholder's utility, economic capacity, and degree of linkage with the problem characteristics, respectively. $M$ captures the impact of additional funding on utility and $\varepsilon$ is the error term that captures aspects of utility that are unobservable. According to Reference [59], the

probability of a "yes" answer to paying a particular fee amount is, therefore, the probability that utility with the fee exceeds utility without the fee.

$$P_i(\text{"yes"}) = P\left[u_{1i}(y_i - t_j, X_i, M_1, \varepsilon_{1i}) \geq u_{0i}(y_i, X_i, M_0, \varepsilon_{0i})\right] \tag{A4}$$

This probability can be estimated using a binary reply model such as a probit or logit, according to Reference [59], by specifying a functional form for the utility function, including the nature of the error term. First, we accept that utility is linear in the payment amount, $t$, and stakeholder features, $X$.

$$u_i(t_j, X_i, M) = \beta_0 + \beta_1 t_j + \sum \beta X_i + \varepsilon_i \tag{A5}$$

The difference between the logit and probit specifications pertains to the hypothesis regarding the distribution of the error term, $\varepsilon_i$ [59]. In the case of a logit specification,

$\varepsilon_i$ is supposed to follow a logistic distribution. Therefore, the probability of a "yes" answer is then given by the equation below.

$$P_i(\text{"yes"}) = \frac{e^{\beta_0 + \beta_1 t_j + \sum \beta X_i}}{1 + e^{\beta_0 + \beta_1 t_j + \sum \beta X_i}} \tag{A6}$$

In the probit specification, the error term $\varepsilon_i$ follows a standard normal distribution between References [0,1], and the probability of a "yes" answer is identified as the equation below.

$$P_i(\text{"yes"}) = 1 - \psi\left[\beta_0 + \beta_1 t_j + \sum \beta X_i\right] \tag{A7}$$

where $\psi$ is the standard normal cumulative distribution function. If we further adopt that WTP is at least zero, mean WTP can be calculated by using the equation below.

$$ean\ WTP = \frac{\beta_0 + \sum \beta X_i \overline{X}}{\beta_1} \tag{A8}$$

where $\overline{X}$ is the mean value of the associated stakeholder characteristic(s), and $\beta_1$ is the coefficient on the payment variable.

**Appendix C.**

**Table A2.** Parameters of priority function and models implemented in $\Omega_{ij}$ matrix for the analysis of the MSPION process.

| Stakeholders Selected for the $\Omega_{ij}$ Optimization Management Matrix: 26 (862 Participants Filled out Forms) | |
|---|---|
| - Agricultural associations included in areas 1, 2, and 3 of the new regulatory framework | - Tourist companies associations included in area 1 of the new regulatory framework. |
| - Local, regional, and state administrations with competences in environment, urban planning, coasts, mining, tourism, and agriculture. | - Ports, nautical association companies of the Mar Menor, and other sectors with a legitimized economical interest or impact within the new regulatory framework. |

| Variable Parameters of the Design: 5 |
|---|
| - % of private stakeholder participation in short-term main investments $\chi_1$<br>- % of private stakeholder participation in long/medium-term main investments $\chi_2$<br>- % of private stakeholder participation in short-term secondary investments $\chi_3$<br>- % of private stakeholder participation in long/medium-term secondary investments $\chi_4$<br>- % of private stakeholder participation in long-term maintenance of solutions implemented $\chi_5$ |

| Models of PPP Configuration: 4 | | | | | |
|---|---|---|---|---|---|
| model 1 | $\chi_1 = 0$ | $\chi_2 = 0$ | $\chi_3 = 0$ | $\chi_4 = 0.25$ | $\chi_5 = 0.25$ |
| model 2 | $\chi_1 = 0$ | $\chi_2 = 0.25$ | $\chi_3 = 0.25$ | $\chi_4 = 0.5$ | $\chi_5 = 0.5$ |
| model 3 | $\chi_1 = 0.25$ | $\chi_2 = 0.5$ | $\chi_3 = 0.75$ | $\chi_4 = 0.75$ | $\chi_5 = 0.75$ |
| model 4 | $\chi_1 = 0.5$ | $\chi_2 = 0.75$ | $\chi_3 = 1$ | $\chi_4 = 1$ | $\chi_5 = 1$ |

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
