# Peer review of "New Strategies to Improve Co-Management in Enclosed Coastal Seas and Wetlands Subjected to Complex Environments: Socio-Economic Analysis Applied to an International Recovery Success Case Study after an Environmental Crisis"

_sustainability, doi:10.3390/su11041039_

Reviewer 1 Report

Dear author,

this is an interesting study with important and intersting results- however, in its present form, I cannot recommend a publication in Sustainability. My major concerns are that the manuscript is too long, several sections are too specific and detailed and might not be of interest for a broad readership, the manuscript also needs a better structure and should include more literature and  more comparisons to other studys/areas.

In detail, I recommend the following:

- restructure the introduction: start with more general informations about coastal seas and coastal wetlands (here you should also include some ecosystems services they provide), when refering to anthropogenic impacts you directly start with "what has caused the pollution" but in the same sentence you refer to "anthropogenic transformation process". Here you should decide if you want to focus only on pullution or if you want to give first some information about other environmental problem. I recommend that you make clear to the reader which general environmental problems exits and what are the consequense for the enviroment, and they provide more specific information on pollution and its consequences. In your section about economic costs it is not clear to what this refers and then you mention restoration in a subclause. Here you should make clear what are the management options and restoration possibility for the environmental problems mentioned before (and what are strategies that exits so far).

- what is singular flora and fauna?

- in its present form, the introduction provides much information but does not show which scientific gaps exists and why your study was neccessary (or why new strategies are neccessary). You should include much more scientific literature and make clear in which fields research is neccessary

- the last part of the introduction (the Mar Menor case study) should be shortend by at least the half. Some of the picture should be moved to the supplementary material. Maybe it whould be also better to include only a small passage of the Mar Menor case study in the indroduction and move the rest to the method section (Study area)

- the method section is also too long and provides too much detailed informtion which might be only of interest for some readers. I therefore suggeest to reduce the method section by 30-40% and move detailed information to the supplementary material (e.g. detailed information from p. 12 l. 349 to p. 13, l. 400

- again in the result section you provide to much information and don´t come to the point. You should think about what is of most interest for all readers and what can they take home from your study

- table 2: you should always use small letters- it is not clear why some words have capital letters

- table 4 is not formatted very well, also this table could be moved to the supplementary material

- the discussion has the same problem as the introduction, it needs a better structure and a clear focus. You start with a very general sentence "the work carried out demonstrates the importance of three common issues in situations of environmental crisis in territories..." So which common issues?? which environmal crisis? (this refers to your study, so why not directly mention the exact problem?) and in which territory?

- in comparison the rest of the manuscript, the discussion is very short and does not provide clear informations. Most important, I suggest to include more scientic research and you should compare your results with other area with similar problemes. The discussion is too focused on your study case.

- I suggest to end the discussion with clear recommendations- highlight the "new strategies to improve co-mangement..." and make clear how these strategies could be transferred to other areas with similar problems. This is absolutely not clear in the moment!

Author Response

First, the author wishes to thank the reviewers for their constructive suggestions and comments about the article presented. The author has endeavored to consider these contributions in a revised version of the manuscript. I believe that the work of the reviewers has contributed significantly to enrich the content, thus improving the quality of the article and making the research more rigorous. I trust that this new version has managed to meet the requested enhancements so as to be worthy of being published in Sustainability Journal.

The new version of the manuscript has also been checked by a native English speaker. Changes have been highlighted in red in the text and answers to comments and suggestions made by the reviewers are detailed below:

Reviewer #1:

Dear author,

this is an interesting study with important and intersting results- however, in its present form, I cannot recommend a publication in Sustainability. My major concerns are that the manuscript is too long, several sections are too specific and detailed and might not be of interest for a broad readership, the manuscript also needs a better structure and should include more literature and  more comparisons to other studys/areas.

 Answer: The author understands the concerns of the reviewer and has tried to produce a shorter and more concise document, focusing only on the topics that may be of interest for a broad readership (the text strictly concerning the article has been reduced from 24 to 21 pages, despite the requests of both reviewers to extend different parts). In addition, new appendix and supplementary material sections have been created to move non-essential information for readers (Figures have been reduced from 12 to 8). The structure has also been improved following the different suggestions from both reviewers and the literature has been expanded to include comparisons to other studies/areas as the reviewer proposed.

In detail, I recommend the following:

 - restructure the introduction: start with more general informations about coastal seas and coastal wetlands (here you should also include some ecosystems services they provide), when refering to anthropogenic impacts you directly start with "what has caused the pollution" but in the same sentence you refer to "anthropogenic transformation process". Here you should decide if you want to focus only on pullution or if you want to give first some information about other environmental problem. I recommend that you make clear to the reader which general environmental problems exits and what are the consequense for the enviroment, and they provide more specific information on pollution and its consequences. In your section about economic costs it is not clear to what this refers and then you mention restoration in a subclause. Here you should make clear what are the management options and restoration possibility for the environmental problems mentioned before (and what are strategies that exits so far).

Answer: The author has proceeded as requested by the reviewer. More general information about coastal seas/ wetlands and the ecosystem services they provide has been included at the beginning of the introduction section. With the sentence cited by the reviewer, the author wanted to introduce differentiating cases where we find specific causes of environmental problems (simplified as “pollution”) from the one affected by diffuse anthropization contexts (indicated through “anthropogenic transformation process", in the case study presented we are facing both situations), but probably the author has not explained this approach adequately. Therefore, the guiding thread of this section has been improved to introduce the problem of diffuse and precise anthropization from a broader perspective. Pollution is essentially one of the many consequences of this anthropogenic process, so the approach to the problem has been refocused to clarify this issue to the reader, as recommended by the reviewer. In addition, in the section on economic costs, this question has also been simplified in order to make it clear what the management options, restoration possibilities for the environmental problems and strategies that exist so far are.

- what is singular flora and fauna?

Answer: The Mar Menor has unique fauna and flora, which make this environment an environmentally valuable place. In the previous version of the article, this issue was only mentioned in a generic way so as not to extend the document even further. In the new version of the article the most representative elements of this ecosystem have been detailed without overly extending the text.

- in its present form, the introduction provides much information but does not show which scientific gaps exists and why your study was neccessary (or why new strategies are neccessary). You should include much more scientific literature and make clear in which fields research is necessary

Answer: Indeed, the author acknowledges this omission and has proceeded as requested by the reviewer. The new version of the introduction focuses clearly on the gaps we found in the current scientific literature and why this case study may be of interest for researchers in this field.

- the last part of the introduction (the Mar Menor case study) should be shortend by at least the half. Some of the picture should be moved to the supplementary material. Maybe it whould be also better to include only a small passage of the Mar Menor case study in the indroduction and move the rest to the method section (Study area)

Answer: The author has proceeded as requested by the reviewer. The last part of the introduction has been greatly reduced. A new section with supplementary material has been created at the end of the document, and four of the pictures have been moved there (a fifth one has been moved to the discussion section). The text now only includes a shorter passage of the Mar Menor, and methodological issues have also been transferred to the method section.

- the method section is also too long and provides too much detailed informtion which might be only of interest for some readers. I therefore suggeest to reduce the method section by 30-40% and move detailed information to the supplementary material (e.g. detailed information from p. 12 l. 349 to p. 13, l. 400

Answer: The author has proceeded as requested by the reviewer. Some parts of the method section such as the mathematical detail of the non-parametric Turnbull and Logit/probit parametric estimations have been moved to an appendix. Other parts have been reduced. In this way, this section has been shortened by almost one third.

- again in the result section you provide to much information and don´t come to the point. You should think about what is of most interest for all readers and what can they take home from your study

Answer: The content of this section has been simplified as far as possible in order to make it more concise, focusing its content on explaining what the most relevant results for international readers of the journal may be. The main elements have been highlighted, and others have been deleted or transferred to an appendix.

- table 2: you should always use small letters- it is not clear why some words have capital letters

Answer: Indeed, the author acknowledges this oversight and has proceeded as requested by the reviewer.

- table 4 is not formatted very well, also this table could be moved to the supplementary material

Answer: The author has proceeded as requested by the reviewer. Table 4 has been moved to a technical appendix and its content and format have been improved.

- the discussion has the same problem as the introduction, it needs a better structure and a clear focus. You start with a very general sentence "the work carried out demonstrates the importance of three common issues in situations of environmental crisis in territories..." So which common issues?? which environmal crisis? (this refers to your study, so why not directly mention the exact problem?) and in which territory?

Answer: The author sought to highlight the relevance of the subject of the case study by introducing the discussion with a general framework of analysis for common issues to environmental crises in coastal lagoons and wetlands. Perhaps this approach had not been clearly explained in the text and could lead readers to confusion. Therefore, the author has tried to clarify this approach and answer the questions addressed by the reviewer in the new version of the manuscript.

- in comparison the rest of the manuscript, the discussion is very short and does not provide clear informations. Most important, I suggest to include more scientic research and you should compare your results with other area with similar problemes. The discussion is too focused on your study case.

Answer: The author has proceeded as requested by the reviewer. Part of the previous results section has been transferred to the discussion and new elements have also been included as suggested by the reviewer in order to compare results with other areas with similar problems.

- I suggest to end the discussion with clear recommendations- highlight the "new strategies to improve co-mangement..." and make clear how these strategies could be transferred to other areas with similar problems. This is absolutely not clear in the moment!

Answer: The author has proceeded as suggested by the reviewer. The concept of “new strategies to improve management” has been highlighted to conclude the discussion. In addition, the question of how these strategies can be exported to other with similar problems in the world has been clearly addressed in the last paragraphs of this section.

Sunday, 27 January, 2019

The Author

Reviewer 2 Report

L. 88: The images are interesting but they are dispensable. Also the explanation of the cases seems too extensive.

L. 122: In order for the reader to dimension the developed urbanism, it would be advisable to give more precise figures, such as the urbanized area with respect to which it is not (in la Manga), hotel beds, number of buildings ...

L. 166:  We do not dispute that the GIS participatory is an innovative technique in the Mar Menor but it has been developed in different cases of study of the world and applied in very different themes. It is convenient to review this point in depth.

L. 471: The maps should be more spaces between them and, in addition, they should contain a symbology of the information shown and a graphic scale.

L. 569: The same as in the previous case. The letters on the map are not very readable.

I missed a more scientific explanation on why the lagoon reduced its level of turbidity; the author alludes to this phenomenon under the effect of surprise. 

I consider that, in the methodological explanation, it is worth an effort to be clearer in the explanation; maybe be less specific in explaining some parameters and more explanatory in the concepts.

Author Response

First, the author wishes to thank the reviewers for their constructive suggestions and comments about the article presented. The author has endeavored to consider these contributions in a revised version of the manuscript. I believe that the work of the reviewers has contributed significantly to enrich the content, thus improving the quality of the article and making the research more rigorous. I trust that this new version has managed to meet the requested enhancements so as to be worthy of being published in Sustainability Journal.

The new version of the manuscript has also been checked by a native English speaker. Changes have been highlighted in red in the text and answers to comments and suggestions made by the reviewers are detailed below:

Reviewer #2:

L. 88: The images are interesting but they are dispensable. Also the explanation of the cases seems too extensive.

Answer: The author has proceeded as requested by the reviewer. The figures have been transferred to a graphic dossier included as supplementary material and the explanation of both cases has been shortened.

L. 122: In order for the reader to dimension the developed urbanism, it would be advisable to give more precise figures, such as the urbanized area with respect to which it is not (in la Manga), hotel beds, number of buildings ...

Answer: The author has proceeded as requested by the reviewer. Some data about urban development in the area have been incorporated to more clearly illustrate the anthropic process of the territory numerically.

L. 166:  We do not dispute that the GIS participatory is an innovative technique in the Mar Menor but it has been developed in different cases of study of the world and applied in very different themes. It is convenient to review this point in depth.

Answer: Indeed, the author acknowledges this issue and has proceeded to demonstrate the relevance of the approach of this case study in the global context of international GIS participatory proposals. This question has been specifically addressed in the introduction section.

L. 471: The maps should be more spaces between them and, in addition, they should contain a symbology of the information shown and a graphic scale.

Answer: The author has proceeded as requested by the reviewer as far as possible (the idea was to fit the set of images into a single image to make the reader understand the complete process carried out for the diagnosis of the scope of action and the areas of preferential works in an easy and very visual way). Detailed information of the maps of this figure (e.g. geologic soil types, hydrogeological legend, specific land uses, hierarchization of wadis, etc.) was not found to be relevant in making the reader understand the main phenomenon and would introduce a great deal of information that may lead the reader to confusion. On the other hand, basic symbols of the information shown and a graphic scale have been more clearly incorporated as suggested by the reviewer (only once given that the maps are understood as a single figure, and repeating them unnecessarily may contribute to obstruct part of the information in some cases).

L. 569: The same as in the previous case. The letters on the map are not very readable.

Answer: Indeed, the author acknowledges this issue here and has proceeded to eliminate unnecessary and non-readable data and to more clearly detail symbols of the information shown and a graphic scale of the map.

I missed a more scientific explanation on why the lagoon reduced its level of turbidity; the author alludes to this phenomenon under the effect of surprise.

Answer: Indeed, the author acknowledges this lack and has proceeded to solve this oversight as requested by the reviewer. The beginning of the recovery of the lagoon has been unexpectedly fast, nevertheless its causes seem clear, since they have been due fundamentally to the cessation of the discharges (legal and illegal) coming from intensive agriculture and the little regulation that this activity had in the area. All this matter is derived from the multidisciplinary diagnosis made and the social pressure that this process fundamentally placed on the agricultural activities, because until now the focus had been placed only on the urbanization activity. This aspect, which as the reviewer rightly points out, had been understood to have been neglected by the author, has now been incorporated in greater detail into the new text. The discussion section has been consequently extended in order to incorporate a wider explanation of this question.

I consider that, in the methodological explanation, it is worth an effort to be clearer in the explanation; maybe be less specific in explaining some parameters and more explanatory in the concepts.

Answer: The author has proceeded as requested by the reviewer and has modified the methodological section to improve its explanation and make it clearer. Some non-essential parts of the method section, such as the mathematical detail of the non-parametric Turnbull and Logit/probit parametric estimations, have been moved to an appendix to reduce parameters explanation and focus better the main concepts of the global process.

Sunday, 27 January, 2019

The Author

Round  2

Reviewer 1 Report

Dear author,

the manuscript has improved a lot after the last revision, I therefore have only two very small suggestions:

- l. 658 tuberculata should be with small letter

- fig. 8: scientific names should be italic, if there is "no seagrass" what do I have instead? sand?

Author Response

The author wishes to thank again the reviewer for his/her constructive suggestions and comments about the article presented. The author has endeavored to consider these contributions in a revised version of the manuscript. I trust that this new version has managed to meet the requested enhancements so as to be worthy of being published in Sustainability Journal.

The new version of the manuscript has been checked again by a native English speaker. New changes have been highlighted in green in the text and answers to comments and suggestions made by the reviewer #1 are detailed below:

Reviewer #1:

Dear author,

the manuscript has improved a lot after the last revision, 

ANSWER: The author thanks the reviewer for the considerate comments.

I therefore have only two very small suggestions:

- l. 658 tuberculata should be with small letter

ANSWER: The author has proceeded as requested by the reviewer.

- fig. 8: scientific names should be italic, if there is "no seagrass" what do I have instead? sand?

ANSWER: The author has proceeded as requested by the reviewer. Indeed, as can be seen (with some difficulty due to the turbidity of the water) in the third photo of figure 8, the death of the seagrass left the seabed only covered with sand.

Tuesday, 5 February, 2019

The Author